# Peer review of "Enhancing Nutrition Communication in Early Childhood Education Settings: Overcoming Challenges and Promoting Collaboration with Caregivers"

_ijerph, 2025, doi:10.3390/ijerph22050677_

Round 1
Reviewer 1 Report
Comments and Suggestions for Authors
The study uses a qualitative approach to examine the issue, however, during the analysis, it prefers the analytical methods used in quantitative research. I find the presentation of relative frequencies problematic due to the low number of elements. The data in the EHS column belonging to the Race/Ethnicity rows in Table 1 cannot be interpreted. In my opinion, a qualitative analysis of the interviews is not done, the authors are actually sharing their own thoughts.
Author Response
|
Comments 1: The study uses a qualitative approach to examine the issue, however, during the analysis, it prefers the analytical methods used in quantitative research. I find the presentation of relative frequencies problematic due to the low number of elements. The data in the EHS column belonging to the Race/Ethnicity rows in Table 1 cannot be interpreted. In my opinion, a qualitative analysis of the interviews is not done, the authors are actually sharing their own thoughts.
Response 1: Note: For your reference, we have used line numbers in Simple Markup in our responses. |
- Thank you for your feedback. Table 1 presents the demographic characteristics of the participants. As outlined in the Methods section, demographic information was collected via Qualtrics (lines 142-145) and analyzed using descriptive statistics in SPSS (lines 168-169). Given that demographic characteristics are typically reported quantitatively, we followed standard analytical approaches for summarizing these data. Thus, we would appreciate clarification regarding the specific concern raised in this comment.
- Thank you for catching that error. There was an issue in the EHS column under the Race/Ethnicity rows in Table 1. We have corrected it by ensuring that percentages are now properly displayed in parentheses.
- Thank you for your feedback. We have revised the Methods section to provide greater clarity on our qualitative analysis approach (Lines 168-183). However, we respectfully disagree with this comment, as our analysis followed standard qualitative research procedures, incorporating both content and thematic analysis. The coding process was conducted systematically, ensuring that themes emerged directly from the data rather than reflecting the authors’ interpretations. Our approach aligns with established qualitative methodologies, enhancing the rigor and trustworthiness of our findings.
Reviewer 2 Report
Comments and Suggestions for Authors
This study is interested, and the selected topic is quite common to be studied. However, some points need to be improved, especially in the materials and methods section.

Author Response
|
Comments 1: This study is interested, and the selected topic is quite common to be studied. However, some points need to be improved, especially in the materials and methods section. · Abstract: o Please define in how this data being analyzed and what software used to this analysis. o Is this study using thematic analysis? if yes, author can present the results in how many themes been analyzed. · Methods: o Authors need to divide with each sub-title of each material and methods section to be more understandable such as study design, informant recruitment, data collection, interview guideline, data analysis, etc. o Explain about the study design first. o Lines 147-156: How to ensure the validity of these data? Did author take a key informant for reference? · Discussion: o Need to divide in each thematic analysis of the study results.
Response 1: Note: For your reference, we have used line numbers in Simple Markup in our responses. · Abstract: We have refined the abstract based on the reviewer’s feedback to provide greater clarity on the analytical approach. “Using inductive thematic analysis, two researchers independently coded the data, identified themes, and compared similarities and differences between EHS/HS and non-EHS/HS providers through an iterative review process.” (Lines 29-32). o We did not use a software to analyze qualitative data. Manual coding allowed for a deeper engagement with the data, ensuring that emergent themes accurately captured participants' experiences. o The results highlighted themes categorized under barriers and facilitators to nutrition communication. Common barriers included limited time, caregiver resistance to change, and language challenges. EHS providers further identified economic constraints, limited nutrition knowledge among caregivers and providers, and restricted access to technology, whereas non-EHS providers emphasized trust issues and caregiver non-compliance with center policies. Facilitators suggested by EHS providers included trust-building strategies and documenting dietary intake for personalized interactions, while non-EHS providers recommended regular meetings and mobile messaging (Lines 33-40) · Methods: Thank you for your suggestion. We have reformatted the Materials and Methods section to include clear subheadings for each component, such as Study Design, Informant Recruitment, Data Collection, Interview Guideline, and Data Analysis, to enhance clarity and readability (Lines 114-184). Regarding your question about ensuring the validity of the data, as mentioned in lines 163-166, we tested the interview guide with a childcare professional to refine the questions and ensure their clarity and relevance. Additionally, we took care to ensure that the data accurately reflected the participants' experiences and perspectives through the iterative coding process and ongoing discussion among the research team to achieve consensus on the final themes (Lines 173-183). · Discussion: Thank you for your suggestion. We have revised the Discussion section to clearly separate the different themes identified in the study, including barriers, facilitators, and specific factors influencing nutrition communication. This restructuring aims to improve the clarity and coherence of the findings, aligning with the thematic analysis and providing a more organized presentation of the results (253-303). |
|
|
Reviewer 3 Report
Comments and Suggestions for Authors
Dear Authors,
Thank you for this very interesting manuscript. I have provided some minor comments below that I believe could help enhance the clarity and overall quality of your work. Please consider the suggestions bellow as you revise the manuscript.
Best regards,
1) Could you please clarify the context in which nutrition communication occurred between the ECE providers and the caregivers? Understanding the setting or conditions under which these interactions took place would provide valuable context for interpreting the findings. For example, on page 198, it is mentioned that brief interactions occurred during drop-off and pick-up. Could you describe this in more detail and possibly include it earlier in the paper?
2) p 62 In the phrase "exposure to various foods," could you please clarify whether you are referring to energy-dense, nutrient-poor foods (EDNP) or if it encompasses a broader range of food types? This clarification would help to better understand the specific dietary influences in the caregiving environments you are discussing.
3) p 96 Could you please clarify the distinction between EHS and non-EHS in your study? It would be helpful to have a more detailed explanation of the specific characteristics or criteria that differentiate these two groups in the context of your research.
4) p 126 please explain in a more structured manner the content of the guide (number of questions, type of questions etc) also please describe the length of the interview
5) p 129 Were the pilot test results published? If so, please include an appropriate reference. Otherwise, kindly provide a brief description of how the pilot test was conducted, including its objectives, sample, and any key findings that informed the main study.
6) p 148 “The recorded interviews were securely stored “
Could you please clarify how the recorded interviews were securely stored in accordance with GDPR guidelines? Specifically, were any encryption methods or restricted access protocols used to ensure data protection and confidentiality?"
7) p 153 Could you please provide more details about the use of Qualtrics in your study? Specifically, could you include information about the company, and any relevant details about the platform's features or security measures, especially regarding data collection and protection
8) p 105 Please consider the use of subparagraphs or subtitles in the Materials and Methods section to improve clarity and organization. For instance, you might want to separate the section into specific subheadings such as 'Subject Recruitment,' 'Inclusion/Exclusion Criteria,' 'Data Collection Procedures,' 'Statistical Analysis,' etc. This would help the reader follow the methodology more easily and allow for better navigation through the article. Please also add a brief paragraph explaining the coding system used in the content analysis of your data.
9) p 189 Please modify the table to ensure that variables such as age are clearly labeled as 'Age (y)' with the unit of years in parentheses. Additionally, under the table, please include a note stating: 'Values are expressed as mean (SD) or median (25th and 75th percentiles) for continuous variables and as absolute numbers (n) and frequencies (%) for categorical variables.' Please adjust the statement according to your data for consistency and clarity.
10) Please also explain the superscripts and abbreviations used in the table.
11) p. 249 Could you please clarify the coding system presented in Table 2? Specifically, it would be helpful if the abbreviations used could be explained below the table. Providing a brief explanation of each code would enhance the understanding of how the data was categorized and analyzed, allowing the reader to interpret the table more effectively.
12) Page 250 – Discussion Section
I would suggest discussing more explicitly the barriers and strategies for effective nutrition communication that are presented in Table 2. Expanding on these points in the discussion could provide a clearer connection between your findings and their practical implications. Additionally, please consider comparing and discussing your findings in relation to other reported studies in the field. This will strengthen the discussion and provide a clearer context for the relevance of your results.
Author Response
|
1. Could you please clarify the context in which nutrition communication occurred between the ECE providers and the caregivers? Understanding the setting or conditions under which these interactions took place would provide valuable context for interpreting the findings. For example, on page 198, it is mentioned that brief interactions occurred during drop-off and pick-up. Could you describe this in more detail and possibly include it earlier in the paper? 2. P62 In the phrase "exposure to various foods," could you please clarify whether you are referring to energy-dense, nutrient-poor foods (EDNP) or if it encompasses a broader range of food types? This clarification would help to better understand the specific dietary influences in the caregiving environments you are discussing. 3. P96 Could you please clarify the distinction between EHS and non-EHS in your study? It would be helpful to have a more detailed explanation of the specific characteristics or criteria that differentiate these two groups in the context of your research. 4. P126 please explain in a more structured manner the content of the guide (number of questions, type of questions etc) also please describe the length of the interview 5. P129 Were the pilot test results published? If so, please include an appropriate reference. Otherwise, kindly provide a brief description of how the pilot test was conducted, including its objectives, sample, and any key findings that informed the main study. 6. P148 “The recorded interviews were securely stored” Could you please clarify how the recorded interviews were securely stored in accordance with GDPR guidelines? Specifically, were any encryption methods or restricted access protocols used to ensure data protection and confidentiality?" 7. P153 Could you please provide more details about the use of Qualtrics in your study? Specifically, could you include information about the company, and any relevant details about the platform's features or security measures, especially regarding data collection and protection 8. P105 Please consider the use of subparagraphs or subtitles in the Materials and Methods section to improve clarity and organization. For instance, you might want to separate the section into specific subheadings such as 'Subject Recruitment,' 'Inclusion/Exclusion Criteria,' 'Data Collection Procedures,' 'Statistical Analysis,' etc. This would help the reader follow the methodology more easily and allow for better navigation through the article. Please also add a brief paragraph explaining the coding system used in the content analysis of your data. 9. P189 Please modify the table to ensure that variables such as age are clearly labeled as 'Age (y)' with the unit of years in parentheses. Additionally, under the table, please include a note stating: 'Values are expressed as mean (SD) or median (25th and 75th percentiles) for continuous variables and as absolute numbers (n) and frequencies (%) for categorical variables.' Please adjust the statement according to your data for consistency and clarity. 10. Please also explain the superscripts and abbreviations used in the table. 11. P249 Could you please clarify the coding system presented in Table 2? Specifically, it would be helpful if the abbreviations used could be explained below the table. Providing a brief explanation of each code would enhance the understanding of how the data was categorized and analyzed, allowing the reader to interpret the table more effectively. 12. Page 250 – Discussion Section. I would suggest discussing more explicitly the barriers and strategies for effective nutrition communication that are presented in Table 2. Expanding on these points in the discussion could provide a clearer connection between your findings and their practical implications. Additionally, please consider comparing and discussing your findings in relation to other reported studies in the field. This will strengthen the discussion and provide a clearer context for the relevance of your results.
Responses 1. Thank you for your thoughtful feedback. We have added further details on the context of nutrition communication between ECE providers and caregivers in lines 68-74. Specifically, we now elaborate on how these interactions typically occur during drop-off and pick-up, as well as through other communication channels such as newsletters and digital platforms. This additional context helps clarify the conditions under which these exchanges take place and enhances the interpretation of our findings. 2. Thank you for your comment. To clarify, when referring to the influence of caregiving environments on children's food choices, mealtime routines, and exposure to various foods, we are specifically highlighting the impact of healthier, nutrient-rich options that can support the development of positive nutrition habits. We have revised and clarified the language in lines 61-65 to better reflect this focus. 3. Thank you for your comment. We have added further clarification regarding the distinction between EHS and non-EHS programs in lines 100-106. This additional information outlines the specific characteristics and criteria that differentiate these two groups in the context of our research, including the focus on low-income families and federally mandated guidelines in EHS programs. 4. Thank you for your comment. We have restructured the methods section for clarity. You can now find detailed information regarding the content of the guide, including the number and type of questions, as well as the length of the interview, under the "Data Collection and Interview Guidelines" section. We hope this provides a clearer and more structured explanation. (Lines 135-166). 5. Thank you for your comment. This was not a formal pilot test, as it involved feedback from a single childcare professional. To provide clarity, we have removed the word "pilot" and included additional description in lines 163-166. 6. Thank you for your comment. We have revised the Data Collection section to address your concerns. The recorded interviews were securely stored in accordance with General Data Protection Regulation (GDPR) guidelines. Specifically, the audio files were encrypted and stored in a password-protected folder with restricted access. Only authorized personnel had access to the data, ensuring confidentiality and data protection throughout the study (Lines 144-147). 7. Thank you for your comment. In response, we have provided additional details regarding the use of Qualtrics in our study (lines 142-145). Qualtrics is a secure and widely used online survey platform, known for its robust data protection measures. 8. Thank you for your suggestion. We have reformatted the Materials and Methods section to include clear subheadings for each component, such as Study Design, Informant Recruitment, Data Collection, Interview Guideline, and Data Analysis, to enhance clarity and readability. (Lines 113-184) 9. Thank you for your comment. We have modified the table to clearly label variables such as "Age" as "Age (y)" with the unit of years in parentheses. Additionally, we have added the following note under the table for clarity and consistency: Values are expressed as mean (SD) or median (25th and 75th percentiles) for continuous variables and as absolute numbers (n) and frequencies (%) for categorical variables." 10. Thank you for your comment. The superscripts and abbreviations used in the table were a typo. We have corrected this issue and ensured that all relevant variables are properly labeled and explained for clarity. 11. We appreciate your comment. To clarify, we have added a brief explanation of each subtheme in Table 2 to provide a clearer understanding of the coding system used. We do not have any abbreviations besides the participant ID, which appears in parentheses after each quote. 12. Thank you for the suggestion. We have expanded the discussion to more explicitly address the barriers and strategies for effective nutrition communication presented in Table 2. By doing so, we connect our findings to their practical implications. We also discuss how strategies like provider training and fostering trust have been shown to improve communication, supporting the need for targeted, context-specific interventions, as highlighted in other studies (Lines 253-303). |
Round 2
Reviewer 2 Report
Comments and Suggestions for Authors
This version is much improved from the previous version. Authors have revised the recommendation. We decide to accept and go through to the next process.
Reviewer 3 Report
Comments and Suggestions for Authors
Dear Authors,
Thank you for considering my comments and revising the manuscript accordingly. I appreciate your thorough efforts to address the suggested improvements. The revised manuscript is now clear and well-structured.
Best regards